

# Hardware implementation of FPGA-based spiking attention neural network accelerator

Shiyong Geng[1,*], Zhida Wang[1,*], Zhipeng Liu[1], Mengzhao Zhang[1], Xuelong Zhu[2] and Yongping Dan[1]

[1] Institute of Integrated Circuits, Zhongyuan University of Technology, Zhengzhou, China
[2] Modern Education Centre (MEC), Zhongyuan University of Technology, Zhengzhou, China
* These authors contributed equally to this work.

## ABSTRACT

Spiking neural networks (SNNs) are recognized as third-generation neural networks and have garnered significant attention due to their biological plausibility and energy efficiency. To address the resource constraints associated with using field programmable gate arrays (FPGAs) for numerical recognition in SNNs, we proposed a lightweight spiking efficient attention neural network (SeaSNN) accelerator. We designed a simple, four-layer structured network, achieving a recognition accuracy of 93.73% through software testing on the MNIST dataset. To further enhance the model's accuracy, we developed a highly spiking efficient channel attention mechanism (SECA), resulting in a significant performance improvement and an increase in test accuracy to 94.28%. For higher recognition speed, we optimized circuit parallelism by applying techniques such as loop unrolling, loop pipelining, and array partitioning. Finally, SeaSNN was implemented and verified on an FPGA board, achieving an inference speed of 0.000401 seconds per frame and a power efficiency of 0.42 TOPS/W at a frequency of 200 MHz. These results demonstrate that the proposed low-power, high-precision, and fast handwritten digit recognition system is well-suited for handwritten digit recognition tasks.

## INTRODUCTION

Spiking neural networks (SNNs) represent the third generation of neural networks that fundamentally differ from traditional artificial neural networks (ANNs) in their information processing mechanisms. Unlike ANNs that use continuous activation functions, SNNs communicate through discrete, binary spike events that more closely mimic the temporal dynamics of biological neural systems.

SNNs are built upon spiking neuron models that maintain an internal membrane potential $V(t)$. The most widely adopted model is the

Corresponding author
Yongping Dan, 6100@zut.edu.cn

leaky integrate-and-fire (LIF) neuron, where the membrane potential evolves according to the differential equation:

$$\tau \frac{dV}{dt} = -V(t) + I(t) \tag{1}$$

where $\tau$ represents the membrane time constant and $I(t)$ denotes the input current. When $V(t)$ exceeds a predefined threshold $V_{th}$, the neuron generates a spike and resets its potential to a resting state $V_{reset}$.

Information in SNNs is encoded in the precise timing and frequency of spikes rather than continuous numerical values. This temporal coding scheme naturally leads to sparse, event-driven computation where processing occurs only when spikes are present, contrasting sharply with the dense matrix operations characteristic of traditional neural networks.

SNNs inherently process both spatial and temporal information through their membrane dynamics and spike timing dependencies. This capability enables SNNs to capture time-dependent patterns and sequential relationships without requiring explicit recurrent connections, making them particularly advantageous for processing dynamic, time-varying data streams.

The energy efficiency of SNNs stems from several fundamental characteristics: (1) sparse, event-driven computation that consumes energy only when spikes occur; (2) elimination of expensive multiply-accumulate operations in favor of simple accumulate-and-compare operations; (3) natural compatibility with neuromorphic hardware architectures that exploit asynchronous processing; and (4) binary spike communication that reduces data movement and storage requirements.

In recent years, SNNs have gradually become a research hotspot in the field of artificial intelligence (*Yamazaki et al., 2022*), regarded as the 'third-generation neural networks' (*Holmes, Sacchi & Bellazzi, 2004*) that can better reflect the information processing mechanism of the human brain (*Frackowiak et al., 2004*) by simulating the spikes between biological neurons. SNNs exhibit unique neurobiological plausibility through their intrinsic capacity for spatiotemporal integration during both computational processing and learning phases (*Cai et al., 2023*). These neuromorphic architectures process discrete binary spike inputs that demonstrate dynamic spatiotemporal sparsity, a characteristic enabling self-regulated neuronal activation patterns.

This biological fidelity facilitates the substitution of conventional multiply-accumulate operations with energy-efficient additive mechanisms, thereby substantially reducing computational complexity while maintaining temporal pattern recognition capabilities. The resultant energy-optimized computational paradigms have demonstrated particular advantages in embedded neural processing applications, especially in low-power scenarios such as Internet of Things (IoT) devices (*Huh, Cho & Kim, 2017*), embedded systems (*Kaelbling, 1993*), and edge computing environments (*Tran et al., 2017*). The sparsity and event-driven nature of SNNs reduces memory accesses by decreasing the number of operations (*Stuijt et al., 2021*), in contrast to traditional neural networks (*Razi & Athappilly, 2005*) where a large percentage of energy consumption comes from algorithm-related memory accesses.

Field programmable gate arrays (FPGAs) represent an ideal platform for implementing SNN hardware accelerators (*Han et al., 2020*) due to their flexible hardware reconfigurability and parallel computing capabilities (*Brown et al., 2012*; *Kulkarni et al., 2019*; *Bhattacharjee et al., 2024*). FPGAs offer numerous advantages in neural network acceleration, including low latency, high efficiency, and effective power consumption control. However, FPGAs have limited resources, and balancing inference speed, power consumption, and model accuracy with these constraints remains a key challenge.

While GPU architectures demonstrate limitations in harnessing the intrinsic sparsity of SNNs during training phases (*Cong et al., 2018*), their fundamental mismatch with temporal accumulation-dominant computations further restricts efficient implementation. Recent advances in neuromorphic computing have yielded specialized ASIC solutions (*Isik, 2023*), including Intel's Loihi (*Davies et al., 2018*), the SpiNNaker system from Manchester (*Manchester & Loeb, 2017*), and IBM's True North (*Akopyan et al., 2015*). Despite achieving benchmark performance, these fixed-function designs face dual challenges of architectural rigidity against evolving SNN topologies and prohibitive fabrication costs. In contrast, modern FPGA platforms leverage reconfigurable logic fabrics and adaptive memory hierarchies to dynamically reconcile computational sparsity with temporal dependencies.

Recent advances in neuromorphic circuit design have yielded systematic improvements in spatiotemporal signal processing. Researchers have enhanced the biological fidelity of LIF neurons through adaptive membrane threshold modulation and dynamic encoding mechanisms (*Lu & Xu, 2022*). Parallel innovations in hardware-software co-design include sparsity-driven acceleration architectures (*Lien & Chang, 2022*) that implement bitmask-compressed event streams to exploit activation sparsity at synaptic granularity. Complementary approaches have introduced convolution-pooling co-design paradigms (*Liu et al., 2023*) that minimize layer-wise latency through fused spatial-temporal operations, achieving 23% higher resource utilization efficiency than conventional pipelined implementations. Scaling these principles, researchers have developed dual-path spiking convolutional neural network (CNN) frameworks (*Parashar et al., 2017*) featuring hierarchical complexity allocation, where compact micro-networks handle basic pattern recognition while macro-structures process high-dimensional temporal features. Additionally, channel balanced workload prediction techniques (*Chen et al., 2022*) have been proposed to optimize computational resource allocation.

However, significant challenges remain in SNN training methodologies. While spike timing dependent plasticity (STDP) (*Caporale & Dan, 2008*) provides biological plausibility, it suffers from poor accuracy and limited scalability to multi-layer networks (*Putra, Hanif & Shafique, 2022*; *Datta et al., 2022*; *Lu & Xu, 2022*; *Lien & Chang, 2022*). Supervised backpropagation algorithms (*LeCun et al., 1988*) and stochastic gradient descent (*Bottou, 2012*), though achieving high accuracy for large models, face theoretical limitations when applied to discrete, non-differentiable spike functions in SNNs.

While neuromorphic hardware demonstrates ultra-low power profiles during SNN inference phases, existing implementations often improve inference speed through network simplification, parameter reduction, and computational complexity decrease.

Although this approach can enhance inference speed, it frequently leads to decreased recognition accuracy and fails to fully exploit SNN potential. Therefore, improving inference speed while guaranteeing network accuracy and optimizing power consumption under FPGA resource limitations represents an urgent research direction.

To address these challenges, this article proposes SeaSNN, an FPGA-based spiking neural network accelerator that introduces a novel spiking efficient channel attention (SECA) mechanism. This mechanism enhances the network's ability to capture key information by weighting spiking signals, thereby improving recognition accuracy while effectively reducing redundant information propagation and maintaining low power consumption. In addition, to optimize hardware parallelism and inference speed, this article implements various hardware architecture optimizations in the FPGA implementation. Firstly, for loop unrolling techniques are employed to unroll operations requiring multiple iterations into parallel computations to reduce loop control overhead; secondly, for loop pipelining is performed to improve throughput and execution efficiency by executing tasks in different iterations simultaneously (*Luo et al., 2023*). For further optimization, this article also employs loop flattening and array partitioning techniques to spread nested loops and partition data and parameters to enhance parallelism and data processing efficiency.

The combined use of these optimizations enables SeaSNN to achieve significant performance improvements on FPGAs. In terms of experimental validation, this article uses the MNIST handwritten digit recognition dataset, and test results show that SeaSNN achieves an inference accuracy of 94.28% on FPGA, which is a significant improvement compared to 93.73% before optimization. The inference speed is also significantly improved, with a single inference time of only 0.000401 s, while power consumption is controlled within 0.42 TOPS/W. Compared with traditional SNN-based accelerator designs, SeaSNN achieves a better balance of accuracy, speed and energy consumption, indicating that the proposed spiking neural network accelerator has excellent performance in handwritten digit recognition tasks and demonstrates wide potential in practical application scenarios.

In summary, this work provides an efficient implementation scheme for SNN hardware accelerator design with the following main contributions:

- The accuracy of the model and hardware execution efficiency are successfully improved by introducing the spiking efficient channel attention mechanism (SECA).
- Various optimizations are performed on the FPGA hardware architecture. Firstly, for loop unrolling techniques are adopted to unroll operations requiring multiple iterations into parallel computation to reduce loop control overhead; secondly, for loop pipelining is performed to improve hardware throughput and execution efficiency by executing tasks in different iterations simultaneously. For further optimization, this article also employs loop flattening and array partitioning techniques, which spread nested loops and split data and parameters to enhance parallelism and data processing efficiency.
- Experimental results show that the FPGA-based SNN accelerator proposed in this article can achieve high-precision inference tasks with low power consumption, which is

especially suitable for applications in resource-limited embedded systems and edge computing scenarios.

The manuscript's organizational framework proceeds systematically through three principal components. We commence our technical exposition in "Methods" with comprehensive analysis of the accelerator's architectural paradigm, including its operational data pipeline, core component implementations, and neural network infrastructure design. "Results and Discussion" subsequently validates these architectural decisions through empirical verification, demonstrating measurement outcomes and performance benchmarks obtained from our FPGA-based SNN implementation. The culminating "Conclusion" synthesizes key findings while projecting potential trajectories for neuromorphic computing development.

## RELATED WORK

With the development of deep learning (*Schmidhuber, 2015*), SNNs have received much attention in many fields due to their bio-inspired and low-power characteristics, especially in applications such as the Internet of Things (IoT) (*Huh, Cho & Kim, 2017*), embedded systems (*Kaelbling, 1993*), and edge computing (*Shi et al., 2016*). SNNs achieve more energy-efficient information processing mechanisms by mimicking the impulsive communication of biological neurons, and thus have an advantage over traditional CNNs (*O'Shea, 2015*). However, hardware implementations of SNNs face a number of challenges, particularly how to optimize accelerator performance while balancing inference speed, accuracy and hardware power consumption (*Maguire et al., 2007*). This section reviews important recent work on FPGA-based SNN accelerators, attention mechanisms in SNNs, and hardware optimization strategies, specifically analyzing their limitations and how our work addresses these challenges.

Several research teams have made significant progress in implementing SNN hardware accelerators on FPGAs, but existing approaches face critical limitations that our work specifically addresses. *Khodamoradi, Denolf & Kastner (2021)* proposed an FPGA-based accelerator design for SNNs, which employs a spiking accumulation mechanism to reduce redundant transmission of spiking data and optimizes inference speed through resource reuse and hardware pipelining techniques. While this approach achieves high inference speed in low-power environments, it suffers from limited network accuracy improvement and lacks sophisticated feature selection mechanisms. Our SeaSNN addresses this limitation by introducing the SECA mechanism, which enhances accuracy while maintaining the energy efficiency advantages of spike accumulation.

Similarly, *Liu, Yenamachintala & Li (2019)* designed a sparsified storage and transmission mechanism based on spiking data characteristics, significantly improving inference speed and energy efficiency through efficient storage architectures and parallel computation models in FPGAs. However, this work primarily focuses on data movement optimization without addressing the challenge of intelligent feature selection within the sparse spike patterns. Building upon their sparsification insights, our SECA mechanism

adds a crucial layer of intelligent channel weighting that selectively processes the most informative sparse spike patterns, thereby achieving both computational efficiency and enhanced recognition capability.

The resource-constrained problem represents a critical challenge when implementing SNNs in FPGAs, particularly in optimizing network structure and parameters for efficient inference with limited hardware resources. Existing optimization strategies have made valuable contributions but leave important gaps that our approach fills.

To address resource constraints, *Hoefler et al. (2021)* proposed pruning and sparsification methods for deep neural networks, removing redundant connections and neurons to reduce hardware resource consumption and accelerate inference. While effective for general network compression, this approach does not specifically exploit the unique temporal sparsity characteristics of SNNs. Our work leverages this pruning foundation but extends it through SECA's dynamic channel attention, which adaptively identifies and emphasizes the most informative spike channels rather than applying static pruning criteria.

In quantization research, *Wu et al. (2023)* proposed low-bit quantization techniques to represent pulse signals and weights in SNNs as low-bit integers, significantly reducing storage requirements and computational complexity. This quantization approach successfully reduces hardware resource consumption but does not address the challenge of maintaining high accuracy under severe resource constraints. Our SeaSNN incorporates these quantization benefits while introducing SECA to compensate for potential accuracy loss through intelligent feature selection, achieving better accuracy-efficiency trade-offs than quantization alone.

Based on these insights, our work implements efficient parallel computation on FPGA through hardware optimization strategies such as loop flattening and array partitioning, extending beyond existing approaches by combining these techniques with attention-driven computation scheduling.

The attention mechanism has become crucial for improving neural network performance (*Niu, Zhong & Yu, 2021*) by dynamically weighting different network components to focus on key feature information. However, the integration of attention mechanisms with SNNs remains significantly underdeveloped compared to their widespread adoption in traditional CNNs, presenting both challenges and opportunities that our work addresses.

Recent pioneering efforts have begun exploring attention in SNNs. *Zhu et al. (2024)* proposed temporal attention-based SNNs that introduce weighting mechanisms in the temporal domain, enabling dynamic adjustment of network connection weights according to time-dimensional spiking information. While this approach demonstrates performance improvements in temporal data processing, it suffers from high implementation complexity and substantial hardware resource demands, making it impractical for resource-constrained FPGA implementations. Our SECA mechanism addresses this limitation by focusing on channel-wise attention rather than complex temporal dependencies, achieving comparable performance benefits with significantly lower computational overhead.

Building on this foundation, *Cai et al. (2023)* proposed a lightweight channel attention mechanism specifically for weighting spiking data in SNNs, demonstrating improved computational efficiency and accuracy through optimized channel distribution of spiking data. However, this work lacks comprehensive hardware implementation validation and does not provide detailed analysis of the attention mechanism's integration with FPGA-specific optimizations. Our SECA design extends their channel attention concept by providing a complete hardware-software co-design solution, demonstrating concrete FPGA implementation results and achieving measurable accuracy improvements (from 93.73% to 94.28% on MNIST).

The reviewed literature reveals three critical gaps that our SeaSNN architecture specifically addresses: (1) existing FPGA-based SNN accelerators achieve speed and efficiency but lack sophisticated accuracy enhancement mechanisms; (2) current resource optimization strategies do not fully exploit SNN-specific sparsity characteristics for intelligent feature selection; and (3) attention mechanisms for SNNs remain either too complex for practical hardware implementation or lack comprehensive validation in resource-constrained environments.

Our work synthesizes the strengths of existing approaches while addressing their limitations through: leveraging proven sparsification and quantization techniques from *Liu, Yenamachintala & Li (2019)*, *Wu et al. (2023)* while adding SECA's intelligent channel selection; building upon the hardware optimization foundations of *Khodamoradi, Denolf & Kastner (2021)*, *Hoefler et al. (2021)* while introducing attention-driven computation scheduling; and extending the channel attention concepts from *Cai et al. (2023)* with complete FPGA implementation and validation.

This comprehensive approach enables SeaSNN to achieve superior accuracy-efficiency trade-offs compared to existing solutions, demonstrating the practical viability of attention-enhanced SNNs in resource-constrained hardware environments.

## METHODS

The system architecture for handwritten digit recognition is shown in Fig. 1. It consists of two main components: a programmable logic module (PL) and a processing system (PS), based on the ZYNQ architecture. The PL includes the SECA module, multi-layer fully connected (FC) layers, and spiking neuron layers (LIF) for feature extraction and spiking signal processing. The FC and LIF layers are alternately arranged and communicate with the PS *via* the AXI interconnection interface (AXI Master).

The PS integrates dual ARM Cortex-A9 CPUs for loading model parameters, displaying results, and controlling system processes. DDR DRAM stores large datasets and model parameters, while high-speed data transmission between the PS and PL is achieved *via* the AXI bus, ensuring efficient computation and data management.

Figure 2 shows the workflow of the handwritten digit recognition system. After initialization, weight parameters are stored on an SD card, read through an interface function, and cached in DDR memory on the PS side. Image data is defined in a RAM array on the PS side. The PL performs computations, transferring image data to DRAM *via*

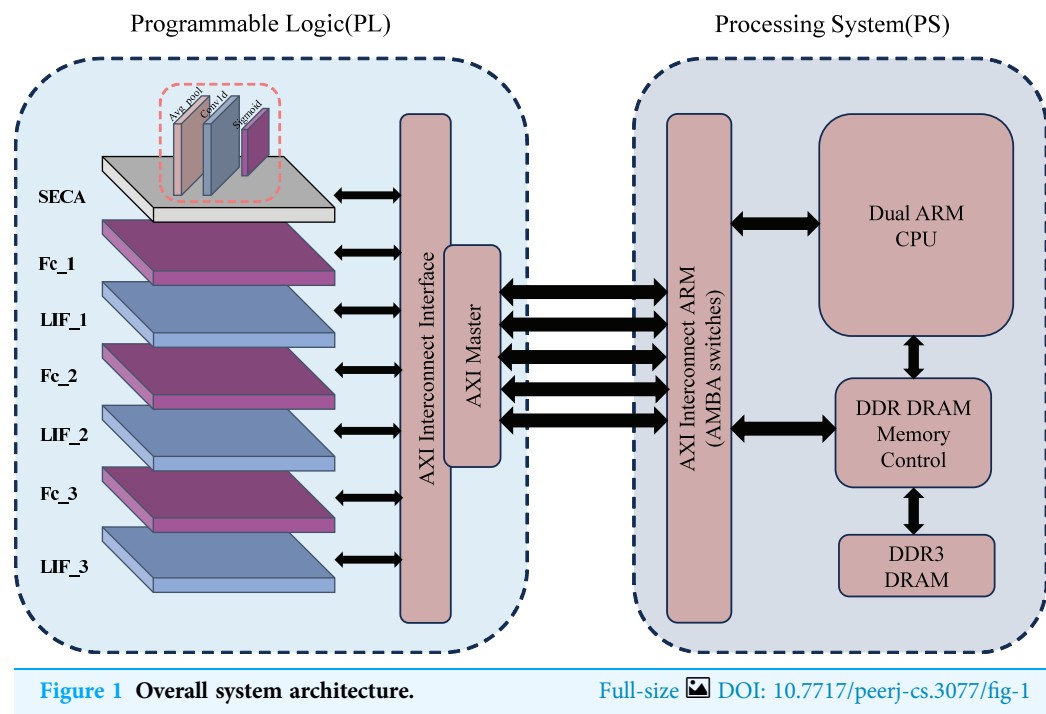

Programmable Logic(PL)    Processing System(PS)

**Figure 1** **Overall system architecture.**

DMA. The ARM processor then sends parameters and image data to the FPGA-based accelerator on the PL for inference, with results returned to the PS.

## Dataset encoding

Encoding the input dataset with spikes offers several advantages: voltage bursts are reduced to discrete single-bit events (1 or 0), simplifying hardware implementation compared to high-precision processing. Sparse activations multiplied by synaptic weights eliminate the need to read most network parameters from memory when values are multiplied by 0, enabling highly efficient hardware computation. This design employs rate coding, where each MNIST sample is repeated over num_steps. Pixel values are mapped to spike frequencies: black (input = 0) generates no spikes, gray (input = 0.5) generates one spike every two time steps, and white (input = 1) generates a spike at every time step.

## LIF neurons

The leaky integrate-and-fire (LIF) neuron model bridges the gap between the Hodgkin-Huxley and artificial neuron models. Similar to an artificial neuron, it computes a weighted sum of inputs but integrates the input over time with a leakage mechanism, resembling an RC circuit. When the integrated value surpasses a threshold, the LIF neuron generates a voltage spike.

LIF neurons simplify output spikes by treating them as discrete events, encoding information in spike timing or frequency rather than shape. This abstraction makes them suitable for studying neural codes, memory, network dynamics, and deep learning. LIF

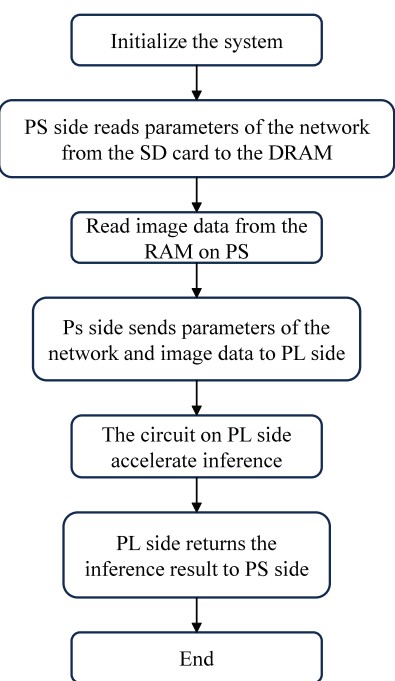

**Figure 2 Handwriting digital literacy system workflow.**

neurons balance biological plausibility and computational efficiency, providing both realism and practicality, as shown in Fig. 3. A neuron fires and resets its membrane potential when $V$ exceeds $V_t hr$, unless in the refractory period $T_R$, where it stays inactive despite surpassing $V_T$. Due to leakage, the membrane potential also decreases without input spikes.

At time step $a$, the variables $U_a^b(i)$ and $S_a^b(i)$ denote the membrane potential and spike of neuron $i$ in layer $b$, respectively. The temporal dynamics are determined by the membrane potential and spike from the previous time step, along with the leakage constant $\alpha$. The spatial dynamics, on the other hand, are shaped by the weighted spikes received from the preceding layer. The firing function, as defined in Eq. (2), is triggered when the membrane potential exceeds the threshold $th_f$, causing the neuron to fire a spike and reset its potential to zero.

$$\begin{cases} U_a^b(i) = \underbrace{\alpha U_{a-1}^b(i)\big(1 - S_{a-1}^b(i)\big)}_{\text{temporal}} + \underbrace{\sum_j S_a^{b-1}(j)w^{b-1}(j,i)}_{\text{spatial}}, \\ S_a^b(i) = fire\big(U_a^b(i) - th_f\big). \end{cases} \quad (2)$$

$$fire(x) = \begin{cases} x, & x \geq 0 \\ 0, & x < 0 \end{cases} \quad (3)$$

Equation (3) illustrates the output sparsity of a neuron during the backward pass. When the membrane potential lies within the threshold range $th_b - th_h$, the value of $\sim S_a^b(i)$ equals 1. Conversely, if the membrane potential falls outside this range, $\sim S_a^b(i)$ is set to 0. When $\sim S_a^b(i)$ is 0, the term $\Delta S_a^b(i)$ is excluded from the update of $\Delta U_a^b(i)$.

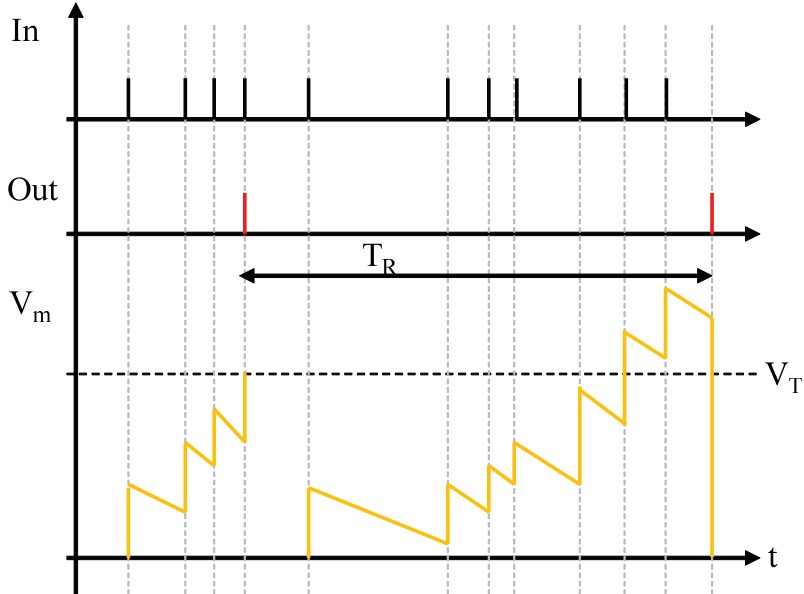

**Figure 3 LIF neurons.**

$$\sim S_a^b(i) \approx \begin{cases} 0, & \text{other} \\ 1, & th_l < u_t^l[i] < th_h \end{cases} \tag{4}$$

$$\begin{cases} \Delta S_a^b(i) = \underbrace{\Delta U_{a+1}^b(i)\left(-\alpha U_a^b(i)\right)}_{\text{temporal}} + \underbrace{\sum_j \Delta U_{a+1}^b(j) w^b(i,j)}_{\text{spatial}}, \\ \Delta U_a^b(i) = \Delta U_{a+1}^b(i)\alpha\left(1 - S_a^b(i)\right) + \Delta S_a^b(i) \sim S_a^b(i). \end{cases} \tag{5}$$

The computation of $\Delta S_a^b(i)$, as described in Eq. (4), involves two components. The temporal component is influenced by the gradient of the potential at the subsequent time step, the membrane potential at that step, and the leakage constant $\alpha$. The spatial component is determined by the weighted sum of the potential gradients from the next layer at that time step. The weight update, given by Eq. (5), is obtained by accumulating the product of the potential gradient from the previous layer and the input spike over time.

$$\Delta w^b(i,j) = \sum_a \Delta U_a^{b+1}(j) S_a^b(i). \tag{6}$$

### SeaSNN model for digital recognition

The network model is implemented using snnTorch (*Eshraghian et al., 2023*), a Python library for gradient-based learning of SNNs, and realized with digital circuits. To achieve high recognition accuracy and inference speed, a seven-layer spiking neural network was designed, as shown in Fig. 4. Training combines spike-timing-dependent plasticity (STDP) with gradient-based learning for handwritten digit recognition on the MNIST dataset.

To deploy SNNs on mobile devices for handwritten digit recognition, two key criteria are essential: high recognition accuracy and low complexity, ensuring compatibility with general FPGA implementations. Based on these criteria, we designed a lightweight

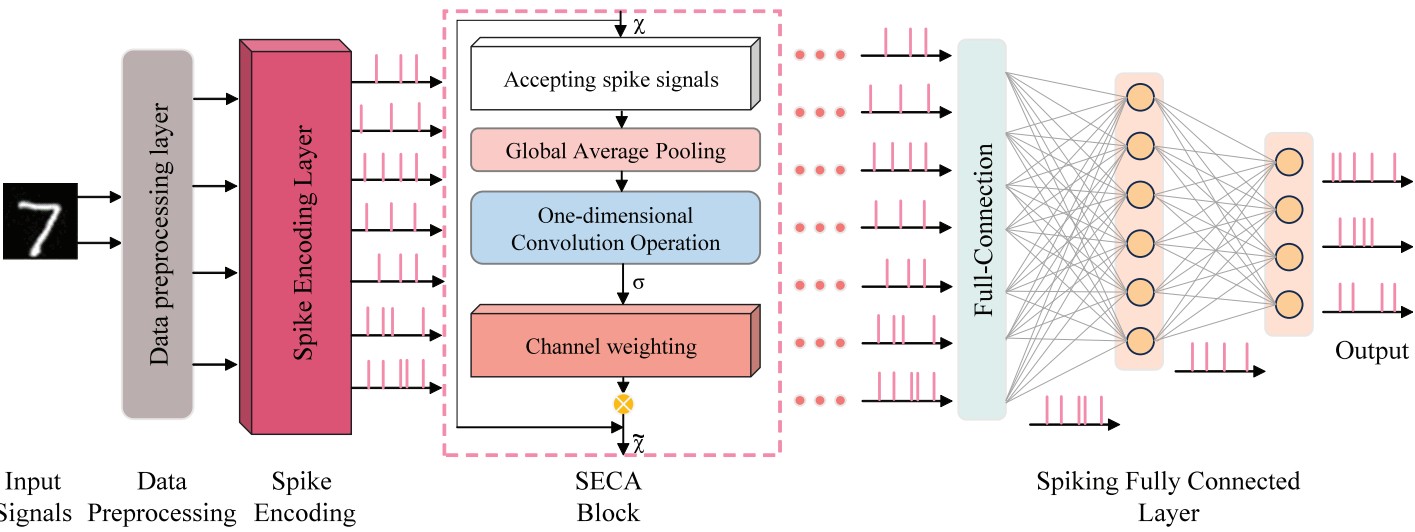

**Figure 4** SeaSNN's lightweight network architecture.

network, "SeaSNN," as shown in Fig. 4. The network comprises seven layers, including a SECA layer, three fully connected layers, and LIF neural layers.

## SECA attention mechanism model

The SECA mechanism captures inter-channel dependencies through local interactions, eliminating the need for additional parameters or complex global pooling operations. A central innovation of SECA is the adoption of a parameter-free one-dimensional local convolution, which facilitates efficient channel-wise interaction while reducing computational overhead and enhancing salient feature identification. The design of the SECA module incorporates insights derived from the analysis of channel dimensionality reduction and cross-channel interactions, as depicted in Fig. 5.

An aggregated feature $\mathbf{y} \in \mathbb{R}^{\mathbb{C}}$, where $C$ denotes the channel dimension, carries channel information through the network.

$$\omega = \sigma(\mathbf{W}\mathbf{y}) \tag{7}$$

Here, $\mathbf{W}$ is a $C \times C$ parameter matrix. A band matrix, $\mathbf{W}_k$, is used to model channel attention and capture local cross-channel interactions.

$$
\begin{bmatrix}
w^{1,1} & \cdots & w^{1,k} & 0 & 0 & \cdots & \cdots & 0 \\
0 & w^{2,2} & \cdots & w^{2,k+1} & 0 & \cdots & \cdots & 0 \\
\vdots & \vdots & \vdots & \vdots & \ddots & \vdots & \vdots & \vdots \\
0 & \cdots & 0 & 0 & \cdots & w^{C,C-k+1} & \cdots & w^{C,C}
\end{bmatrix} \tag{8}
$$

The matrix contains $k \times C$ parameters, as shown in Eq. (7), and is generally small. Equation (8) ensures some interaction between groups, while Eq. (6) captures local dependencies by focusing on interactions between $y_i$ and its $k$ neighbors. This method strikes a balance between preserving local context and computational efficiency.

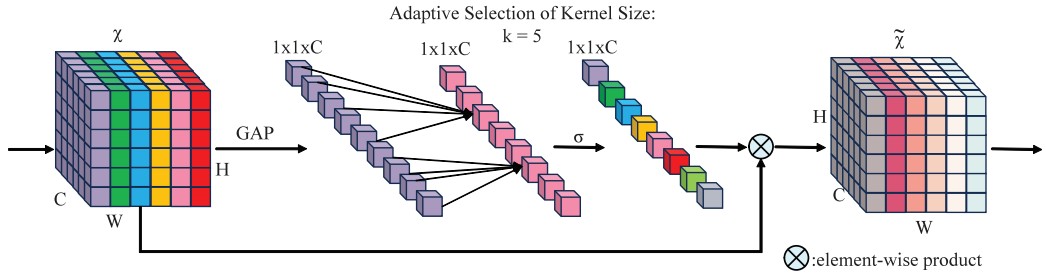

**Figure 5 SECA attention mechanism working principle.**

$$\omega_i = \sigma\left(\sum_{j=1}^{k} w_i^j y_i^j\right), y_i^j \in \Omega_i^k \tag{9}$$

Here, $\Omega_i^k$ denotes the set of $k$ neighboring channels of $y_i$, capturing nearby channels that contribute to local cross-channel interactions.

A more efficient strategy involves having all channels utilize the same learning parameters, *i.e.*:

$$\omega_i = \sigma\left(\sum_{j=1}^{k} w^j y_i^j\right), y_i^j \in \Omega_i^k \tag{10}$$

This strategy can be efficiently implemented using a fast 1D convolution with kernel size $k$, *i.e.*:

$$\omega = \sigma(\text{C1D}_k(\mathbf{y})) \tag{11}$$

Here, C1D denotes a 1D convolution operation. The method in Eq. (5) is implemented *via* the SECA module, using only $k$ parameters. With $k = 3$, it balances model complexity and local cross-channel interaction capture (*Kulkarni et al., 2019*), enhancing efficiency and effectiveness.

Group convolutions have advanced CNN architectures by applying convolutions to channels of varying dimensions, depending on the number of groups. Inspired by this, we propose a correlation between the interaction extent, represented by the kernel size $k$ in 1D convolution, and the channel dimension $C$. This relationship is formalized as a functional mapping, $\varphi$, linking $k$ and $C$:

$$C = \phi(k). \tag{12}$$

The function $\varphi$ defines how the optimal kernel size $k$ adapts to changing channel dimensions, offering a framework to tailor interaction ranges across architectures. This reduces reliance on manual tuning and resource-intensive cross-validation.

The simplest mapping, $\varphi(k) = \gamma k - b$, is inherently limited. Since channel dimensions $C$ are often powers of two, introducing nonlinearity to $\varphi(k)$ improves flexibility and better exploits these characteristics.

$$C = \phi(k) = 2^{(\gamma * k - b)}. \tag{13}$$

Then, given the channel dimension $C$, the suitable kernel size $k$ can be dynamically determined through an adaptive process by:

$$k = \psi(C) = \left| \frac{log_2(C)}{\gamma} + \frac{b}{\gamma} \right|_{odd}. \tag{14}$$

Here, $|t|_{odd}$ denotes the nearest odd number to $t$. In all experiments, $\gamma$ and $b$ are set to 2 and 1, respectively. This mapping ensures that high-dimensional channels interact over larger distances, while low-dimensional channels interact over shorter ones. The nonlinear mapping dynamically adjusts the kernel size according to channel dimensions, improving cross-channel interaction efficiency.

## Circuit optimization method

SeaSNN's acceleration circuitry is implemented using high-level synthesis (HLS), allowing developers to focus on algorithm design rather than circuit implementation, thus improving development efficiency. The modular design maps each module to a SeaSNN layer, storing intermediate feature data in DDR3 DRAM instead of the FPGA. This reduces FPGA storage demands and enhances network scalability and adaptability. Each layer's circuit code is encapsulated as an IP core, which is interconnected in Vivado to build the SeaSNN circuit. The ZYNQ7 processing system connects these IP cores *via* four AXI high-performance (HP) buses, ensuring high-throughput data exchange between the processing system (PS) and programmable logic (PL).

Circuits synthesized with HLS generally require more FPGA resources and longer execution times than those using hardware description languages (HDLs). However, with proper optimizations, HLS-synthesized circuits can achieve comparable performance. To minimize delays in the SeaSNN circuit and enable handwritten digit recognition, techniques like loop unrolling, loop pipelining, and array partitioning are employed.

### Unrolling the for_loop

The schematic of unrolling the for loop (Fig. 6) illustrates a loop with three iterations, each performing a multiplication followed by an addition. Without unrolling, the loop uses a single circuit, executing iterations sequentially. With the UNROLL optimization, three circuits are synthesized to execute iterations in parallel, tripling FPGA resource usage but reducing execution time by 66%. This trade-off significantly accelerates convolution calculations, enhancing inference efficiency.

### Pipelining the for_loop

When a for loop transfers data between the PS and PL, pipelined optimization (PIPELINE) improves execution efficiency. As shown in Fig. 7, the operation "input [i] /= pix;" involves four sequential steps: address calculation, data read, division, and result write-back (Fig. 7B). PIPELINE optimization converts these steps into a 4-stage pipeline, enabling parallel execution (Fig. 7C) and reducing execution time by approximately 75%.

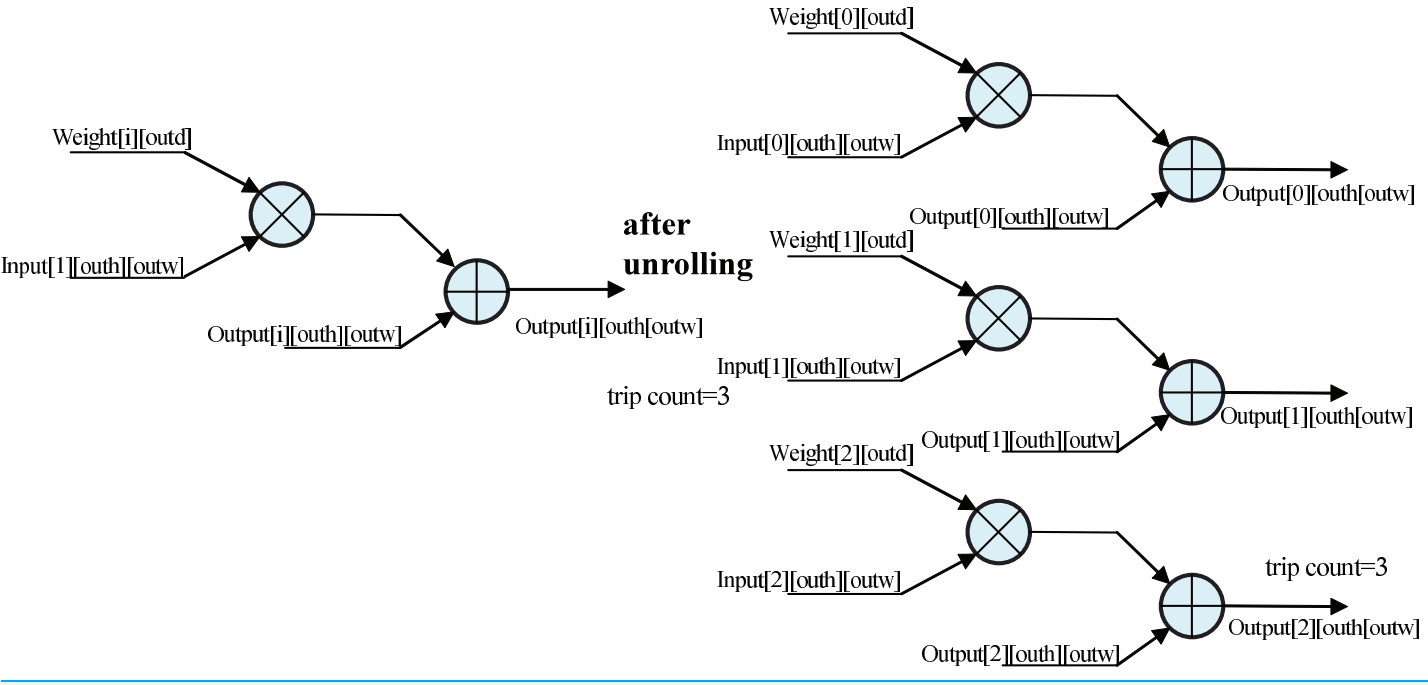

**Figure 6 Unrolling the for_loop.**               

### Array partitioning

Low data access bandwidth is the primary bottleneck in SeaSNN's parallel computing. Limited RAM access ports in FPGA's on-chip RAM lead to conflicts when multiple circuits access the same RAM, restricting parallelism. ARRAY_PARTITION optimization addresses this by splitting data into smaller RAMs, reducing conflicts and enhancing efficiency and throughput. As shown in Fig. 8, methods include block, round-robin, and complete partitioning. Complete partitioning, when resources permit, maximizes throughput and circuit efficiency.

## RESULTS AND DISCUSSION

In this section, we evaluate the proposed SeaSNN model on the MNIST benchmark dataset. All experiments were conducted on a computer equipped with a 3.6 GHz Intel (R) CoreTM i7-12700KF processor and an NVIDIA GeForce GTX 3090 graphics card (24G RAM). The deep learning framework used is PyTorch, and the optimizer is Adam, with a learning rate of 2e-3.

### Effect of neuron number on SeaSNN

The parameter adjustment results of the SeaSNN model are summarized in Table 1. A large number of hidden-layer neurons increases model size and hardware resource consumption, while too few neurons reduce accuracy. To balance accuracy and model size, the final configuration includes 20, 10, and 10 LIF neurons across three hidden layers. After five training epochs, the model achieved 94.28% accuracy with a compact 600 KB size, making it ideal for FPGA implementation.

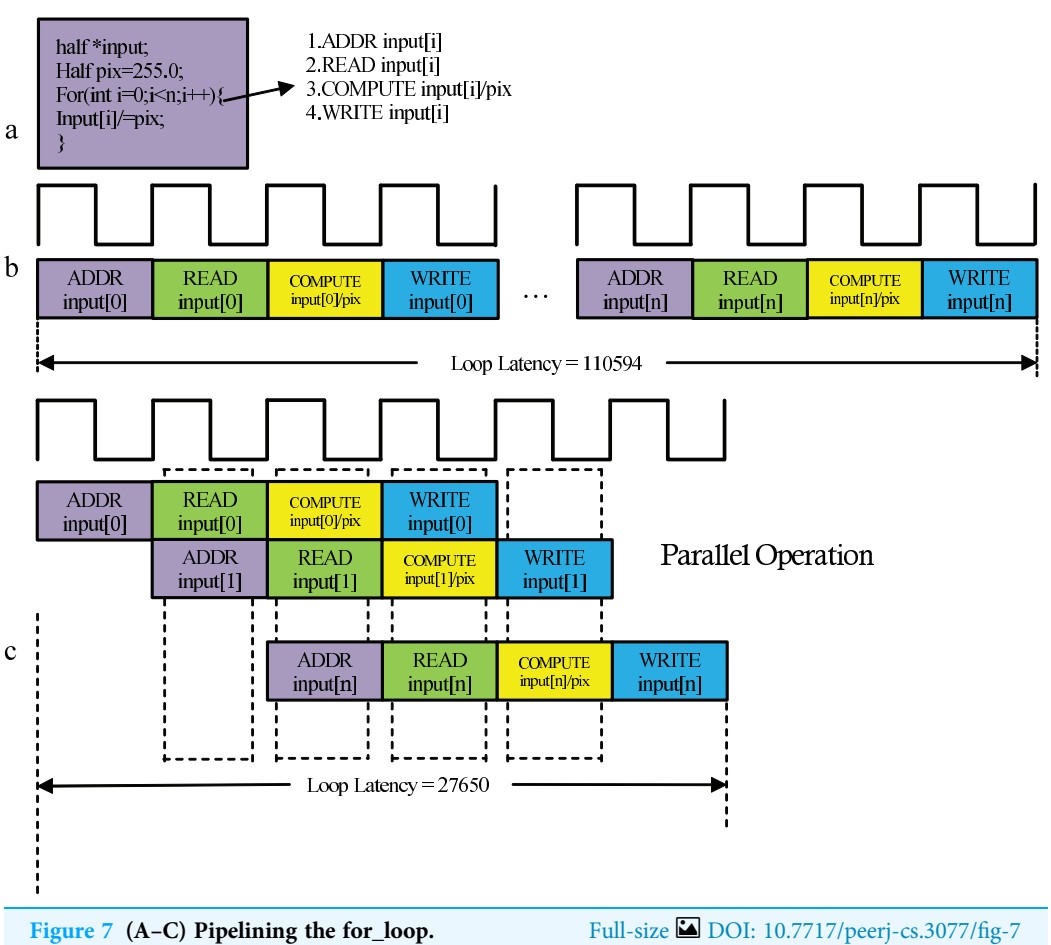

**Figure 7** (A–C) Pipelining the for_loop.

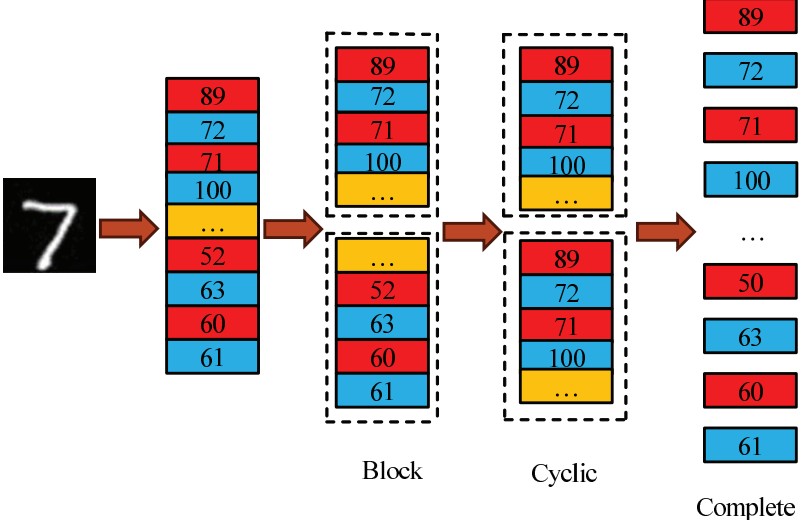

**Figure 8** Array partitioning.

**Table 1 The effect of different numbers of neurons on the network.**

| Epoch | Num_hid1 | Num_hid2 | Num_hid3 | Model size (MB) | Accuracy (%) |
|---|---|---|---|---|---|
| 5 | 300 | 100 | 10 | 1.32 | 96.94 |
| 5 | 150 | 50 | 10 | 0.64 | 96.33 |
| 5 | 75 | 25 | 10 | 0.29 | 95.30 |
| 5 | 20 | 10 | 20 | 0.09 | 93.00 |
| 5 | 10 | 10 | 10 | 0.031 | 82.78 |
| 5 | 20 | 10 | 10 | 0.06 | 94.28 |

**Table 2 The effect of different data types on the circuit.**

| Type of data | BRAM | DSP | FF | LUT | Accuracy (%) |
|---|---|---|---|---|---|
| Float (32 bits) | 6.5 | 223 | 54,195 | 57,873 | 94.28 |
| Half (16 bits) | 4.6 | 192 | 48,838 | 56,903 | 93.69 |

## Data type of SeaSNN acceleration circuit

High-precision data enhances SeaSNN's feature extraction and regression performance but increases FPGA storage demands. If resource requirements exceed FPGA capacity, implementation becomes infeasible. In Table 2, during the training of the 16-bit precision model, we modelled the effect of 16-bit quantisation on the weights and activation values in forward propagation. This was achieved by inserting pseudo-quantisation nodes/operations to quantise values to 16-bit precision and then inverse quantising them back to 32-bit for subsequent operations, whilst calculating the gradient using a pass-through estimator in the backpropagation. This approach allows the model to learn parameters that are robust to reduced precision, thus minimising the accuracy degradation compared to post-training quantisation. To evaluate data type impacts, 32-bit single-precision (float-type) and 16-bit half-precision (half-type) formats were tested. Half-type reduces DSP, FF, and LUT usage but compromises accuracy and increases complexity. Float-type, achieving 94.28% accuracy compared to 93.69% for half-type, ensures superior precision, stability, and robustness, critical for deep learning. It avoids feature blurring, performs better in complex scenarios, and supports numerical stability and convergence. Despite higher resource demands, the float-type's platform compatibility and precision make it optimal for current and future applications.

## The impact of SECA attention module

The SECA attention mechanism efficiently models channel dependencies using 1D convolution (1D Conv) with minimal parameters and computational overhead. In the SeaSNN accelerator, SECA enhances feature extraction in the first layer by emphasizing critical channel information and suppressing irrelevant data, improving feature quality and network efficiency. It accelerates network convergence by stabilizing gradient propagation and reducing redundant computations, enabling faster learning of effective features. SECA also improves generalization by filtering out noise during early feature extraction, enhancing robustness and performance, especially in noisy or complex tasks. In

response to the generalizability of our proposed SECA module, we argue that it has the potential to be applied beyond the MNIST dataset and to a wider range of SNN tasks. The core mechanism of the SECA module lies in explicitly modeling inter-channel dependencies through impulse-based manipulations to dynamically recalibrate the channel-level pulse feature responses. This principle of adaptively enhancing or suppressing feature channels based on information importance is not specific to a simple task, but rather a generalized means for SNN to enhance feature representation capabilities, similar to the channel attention mechanism in traditional ANNs (which has demonstrated broad effectiveness in a wide range of visual tasks). Therefore, we expect the SECA module to be beneficial for more complex SNN applications, such as enhancing model performance by augmenting saliency impulse feature channels in image classification tasks such as CIFAR-10/100, event-based visual data processing (N-MNIST, DVS-CIFAR10), and even when impulse neural networks are applied to tasks such as target detection or segmentation. The SECA module is a lightweight and easy-to-use module. Moreover, the lightweight and plug-and-play design features of the SECA module facilitate its integration into different SNN architectures. While full empirical validation on a wider range of tasks and datasets is an important direction for our future work, the underlying design principles of the SECA module suggest its potential as a general-purpose component for improving SNN performance.

## Optimization results of acceleration circuit

To fully utilize FPGA on-chip resources and accelerate handwritten digit recognition, the SeaSNN circuit was optimized using techniques such as UNROLL, PIPELINE, and ARRAY_PARTITION.

Table 3 summarizes the latency and resource utilization of the SeaSNN circuit before and after optimization. Initially, the circuit has a latency of 1,877,004 cycles and utilizes 37 BRAMs, 128 DSPs, 13,025 FFs, and 17,996 LUTs. After optimization, latency is reduced to 437,342 cycles—just 23.33% of the original. Resource utilization changes to 37 BRAMs, 120 DSPs, 10,718 FFs, and 15,984 LUTs. Although resource consumption increases slightly in some areas, the significant latency reduction enhances overall resource efficiency and accelerates handwritten digit recognition.

## Resource utilization

Figure 9 illustrates the FPGA resource utilization of the SeaSNN accelerator. LUT and FF resources dominate, indicating heavy use of logic units and flip-flops, while BRAM and DSP usage is comparatively lower, reflecting minimal storage and computational resource demands.

## Evaluation of FPGA and other computing devices

We evaluated the power consumption after place-and-route using Vivado Power Analyzer. The total on-chip power is estimated to be 4.996 W, and the system reaches a junction temperature of 36.6 °C. Based on the achieved throughput of 2.11 TOPS, the resulting inference efficiency is 0.42 TOPS/W, demonstrating a favorable trade-off between

**Table 3 Latency and resource utilisation before and after circuit optimisation.**

| Optimization | BRAM | DSP | FF | LUT | Latency (Cycles) |
|---|---|---|---|---|---|
| without | 37 | 120 | 10,178 | 15,984 | 1,877,004 |
| with | 37 | 128 | 13,025 | 17,996 | 437,342 |

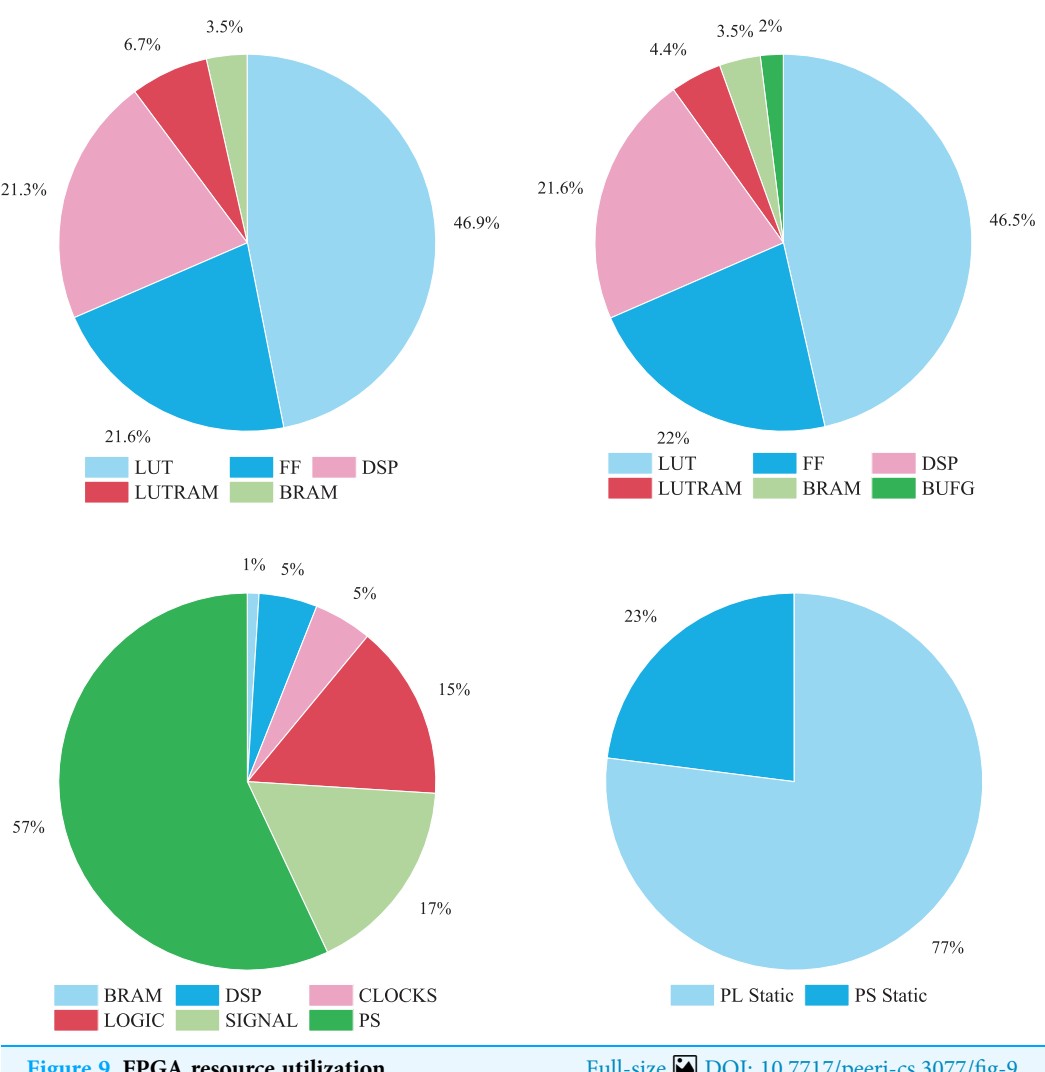

**Figure 9 FPGA resource utilization.**

**Table 4 Comparison of hardware implementations.**

|  | TCAD23 (*Ye, Chen & Liu, 2023*) | TCASI22 (*Liu et al., 2022*) | BioCAS23 (*Wang et al., 2023*) | TCASII24 (*Wang et al., 2024*) | This work |
|---|---|---|---|---|---|
| Platform | Xilinx XC7K325T | Xilinx XC7K325T | Xilinx XCKU115 | Xilinx KC705 | Xilinx XCZU4EF |
| Frequency | 200 MHz | 200 MHz | 120 MHz | 100 MHz | 200 MHz |
| Format | 16 bit | 16 bit | 8 bit | 8 bit | 32 bit |
| LUT | 170,429 | 46,371 | 92,376 | 92,376 | 57,853 |

|  | TCAD23 (*Ye, Chen & Liu, 2023*) | TCASI22 (*Liu et al., 2022*) | BioCAS23 (*Wang et al., 2023*) | TCASII24 (*Wang et al., 2024*) | This work |
|---|---|---|---|---|---|
| FF | 113,138 | 30,417 | 24,707 | 38,243 | 54,915 |
| DSP | 0 | 65 | 0 | 0 | 223 |
| Inference throughput | 3.20 GOPS | 14.76 GOPS | 2.56 GOPS | 3.06 GOPS | 2.11 TOPS |
| Inference efficiency | 6.78 GOPS/W | 27.85 GOPS/W | 1.53 GOPS/W | 14.57 GOPS/W | 0.42 TOPS/W |

performance and energy consumption. Table 4 compares several SNN accelerators proposed in recent years. The results indicate that the proposed SeaSNN accelerator is highly competitive among FPGA-based accelerators. Notably, the accelerator demonstrates significant advantages in terms of throughput and power consumption compared to ASIC-based solutions. Specifically, the proposed design is thousands of times more efficient in power consumption and performance than other designs. These results highlight the superiority of the SeaSNN accelerator in both performance and resource utilization.

## CONCLUSION

This article presents a low-power, high-precision FPGA-based system for accelerating spiking neural networks (SNNs) in handwritten digit recognition tasks. To mitigate accuracy degradation due to hardware constraints in existing SNN accelerators, a simple and efficient four-layer network, "SeaSNN," is proposed, incorporating a pulse channel attention mechanism (SECA) to boost performance. Experiments on the MNIST dataset demonstrate that SECA improves accuracy from 93.73% to 94.28%, validating its effectiveness.

To boost inference speed, FPGA circuit implementation is optimized using loop unrolling, pipelining, and array partitioning, greatly enhancing parallelism. These optimizations achieve an inference speed of 0.000401 seconds per frame at 200 MHz with a power efficiency of 0.42 TOPS/W. The results demonstrate high accuracy and efficiency in handwritten digit recognition, with low power consumption, making the system ideal for resource-constrained edge computing.

This work offers a practical solution for deploying SNNs on hardware and provides insights into network and hardware optimization. Future research will focus on more complex SNN architectures and advanced hardware techniques to further advance SNN adoption in real-world applications.

## ACKNOWLEDGEMENTS

We express our gratitude to the authors of the 'MNIST dataset' for generously providing open data sets to support our research endeavors. Furthermore, we have appropriately referenced the articles authored by the creators of these data sets.

### Funding

This research is supported by the Henan Provincial Science and Technology Research Project (242102210006). The funders had no role in study design, data collection and analysis, decision to publish, or preparation of the manuscript.

### Grant Disclosures

The following grant information was disclosed by the authors:
Henan Provincial Science and Technology Research Project: 242102210006.

### Competing Interests

The authors declare that they have no competing interests.

### Author Contributions

- Shiyong Geng conceived and designed the experiments, performed the experiments, analyzed the data, performed the computation work, prepared figures and/or tables, authored or reviewed drafts of the article, and approved the final draft.
- Zhida Wang conceived and designed the experiments, performed the experiments, analyzed the data, performed the computation work, prepared figures and/or tables, and approved the final draft.
- Zhipeng Liu conceived and designed the experiments, performed the experiments, analyzed the data, prepared figures and/or tables, and approved the final draft.
- Mengzhao Zhang conceived and designed the experiments, performed the experiments, analyzed the data, prepared figures and/or tables, and approved the final draft.
- Xuelong Zhu conceived and designed the experiments, performed the experiments, authored or reviewed drafts of the article, and approved the final draft.
- Yongping Dan conceived and designed the experiments, performed the experiments, prepared figures and/or tables, authored or reviewed drafts of the article, and approved the final draft.

### Data Availability

The original code is available in the Supplemental File.

### Supplemental Information

Supplemental information for this article can be found online at http://dx.doi.org/10.7717/peerj-cs.3077#supplemental-information.

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
