# Peer review of "Hardware implementation of FPGA-based spiking attention neural network accelerator"

_PeerJ Computer Science, doi:10.7717/peerj-cs.3077_

## Round 0.1 · original submission · Minor Revisions

Please pay attention to the reviewer comments when submitting your revision.

**Language Note:** The review process has identified that the English language must be improved. PeerJ can provide language editing services - please contact us at [email protected] for pricing (be sure to provide your manuscript number and title). Alternatively, you should make your own arrangements to improve the language quality and provide details in your response letter. – PeerJ Staff

·

Basic reporting

The manuscript is generally well-written, and the authors clearly communicate their contributions. However, minor grammatical and stylistic improvements would enhance clarity and readability. For instance:

* Line 28: “...have gradually become a research hotspot in the field of Artificial Intelligence(AI)” → Insert space before the parentheses.

* Line 31–33: Sentence repetition between lines 30–33 can be merged for conciseness.

* Several instances of inconsistent spacing and formatting (e.g., missing spaces before citations) should be revised.

Figures are high-quality and well-labeled. The architecture diagrams and workflow figures add value and clarity. All cited data appears to be provided, and references are extensive and relevant.

Experimental design

The study is within scope and presents a meaningful and clearly defined research problem—how to design a high-efficiency FPGA-based accelerator for SNNs under resource constraints.

The SeaSNN design, along with the proposed Spiking Efficient Channel Attention (SECA) module, is well-motivated and rigorously implemented. The methodology for loop optimizations (UNROLL, PIPELINE, ARRAY PARTITION) and their impact on latency and resource usage are described in sufficient technical detail, allowing for reproducibility.

One suggestion: Please clarify whether quantization-aware training was applied when comparing 32-bit vs 16-bit models (Table 2).

Validity of the findings

The results are well-supported with experimental benchmarks. The improvement in accuracy, latency, and power efficiency is clearly demonstrated. The comparison with existing works (Table 4) solidifies the significance of the presented approach.

It would benefit readers if the authors:

* Briefly comment on the potential generalizability of their SECA module to other SNN tasks beyond MNIST.

* Elaborate slightly more on the training setup (e.g., optimizer, learning rate) in Section 4 to reinforce reproducibility.

Reviewer 2 ·

Basic reporting

This paper proposes SeaSNN, an FPGA-based accelerator for spiking attention neural networks. Evaluated on Xilinx FPGAs, SeaSNN achieves high throughput and energy efficiency.
Minor:
Please proofread the manuscript for grammar and punctuation. For example, the sentence "While GPU architectures ... during training phasesCong et al." is missing a comma between "phases" and "Cong et al."

Experimental design

The evaluation is well-defined—for example, Table 4 provides a clear summary comparing area, throughput, and energy efficiency between SeaSNN and prior accelerators. However, what about accuracy? Did previous works also evaluate on the MNIST digit recognition task? Does SeaSNN achieve comparable or superior accuracy?

Validity of the findings

Impact and novelty look good to me.

Additional comments

(1) In the introduction, it would be helpful if the authors could justify the relevance of evaluating SeaSNN on the MNIST handwritten digit recognition task. To my knowledge, compact models such as MobileNet, sparse CNNs, and quantized networks like Binary Neural Networks (BNNs) are also highly efficient for digit recognition. Can SNNs demonstrate superior energy efficiency compared to these customized CNNs on this task? If not, are there other applications where SNNs outperform traditional CNNs in terms of energy efficiency or performance?

(2) At a high level, SNNs encode inputs into spikes, process them over time, and fire only when the accumulated potential exceeds a threshold. This seems conceptually similar to sparse CNNs, which prune less significant activations to reduce computation. What is the core intuition behind SNNs achieving higher energy efficiency than sparse CNNs?

(3) The evaluation uses 32-bit single precision and 16-bit half-precision data types. Can this level of precision offer higher energy efficiency than binary neural networks (BNNs), which typically operate at 1-bit precision? Or even sparse BNN?

Reviewer 3 ·

Basic reporting

The article’s organization is sound, but the writing needs to be improved:

1. The introduction surveys numerous SNN related studies but fails to provide a clear, formal overview of core SNN concepts.

2. The Related Work section catalogs prior research but does not explain how their limitations are addressed or how their strengths inform this article’s contributions.

3. Several typos must be corrected—for example, “SeaSNN” is inconsistently rendered as “SeaSnn,” and in Table III the latency even increased with optimization.

Experimental design

This article provides a clear description of its experiment.

Validity of the findings

The results are not good which raises concerns about the benefit of this design:

1. The inference accuracy (94.28%) based on MNIST dataset is lower than modern SNN benchmarks.

2. The hardware throughput is lower while utilization is higher than other hardware implementations in Table 4.

3. The DSP usage is not really increased after unrolling (if the loop contains multipliers) in Table 3 which needs to be more illustrated.

---

## Round 0.2 · accepted · Accept

The authors have addressed the concerns of the reviewers in this manuscript.